# Barcode-free multiplex plasmid sequencing using Bayesian analysis and nanopore sequencing

**Masaaki Uematsu[1]\*, Jeremy M Baskin[1,2]\***

[1]Weill Institute for Cell and Molecular Biology, Cornell University, Ithaca, United States; [2]Department of Chemistry and Chemical Biology, Cornell University, Ithaca, United States

## eLife Assessment

This study provides an **important** computational tool for analyzing and deconvoluting a pool of plasmids sequenced without barcoding using nanopore long-read sequencing. The tool, which has been **convincingly** validated, is readily available to scientists interested in rapid and cost-effective verification of plasmid sequences as well as in scaling up analysis by pooling samples within barcodes.

**\*For correspondence:**
mu84@cornell.edu (MU);
jeremy.baskin@cornell.edu (JMB)

**Abstract** Plasmid construction is central to life science research, and sequence verification is arguably its costliest step. Long-read sequencing has emerged as a competitor to Sanger sequencing, with the principal benefit that whole plasmids can be sequenced in a single run. Nevertheless, the current cost of nanopore sequencing is still prohibitive for routine sequencing during plasmid construction. We develop a computational approach termed Simple Algorithm for Very Efficient Multiplexing of Oxford Nanopore Experiments for You (SAVEMONEY) that guides researchers to mix multiple plasmids and subsequently computationally de-mixes the resultant sequences. SAVEMONEY defines optimal mixtures in a pre-survey step, and following sequencing, executes a post-analysis workflow involving sequence classification, alignment, and consensus determination. By using Bayesian analysis with prior probability of expected plasmid construction error rate, high-confidence sequences can be obtained for each plasmid in the mixture. Plasmids differing by as little as two bases can be mixed as a single sample for nanopore sequencing, and routine multiplexing of even six plasmids per 180 reads can still maintain high accuracy of consensus sequencing. SAVEMONEY should further democratize whole-plasmid sequencing by nanopore and related technologies, driving down the effective cost of whole-plasmid sequencing to lower than that of a single Sanger sequencing run.

## Introduction

Plasmid construction is a core technique in life science research. Conventionally, it is performed by ligation or assembly of DNA fragments typically generated by PCR amplification or solid-phase oligonucleotide synthesis into a linearized vector (*Chao et al., 2015*). Because errors can be introduced at the PCR amplification, chemical synthesis, ligation, or assembly steps, the final products must be confirmed by a sequencing method. Classically, Sanger sequencing is used, and it involves synthesis of a complementary DNA strand in a polymerase-catalyzed reaction doped with chain-terminating, dideoxynucleotides bearing one of four different fluorescent dyes (*Sanger et al., 1977*; *Smith et al., 1986*). Stochastic chain termination produces a series of fluorescent products whose lengths and fluorescence properties are measured by capillary electrophoresis, allowing the base calling at each

position. Sanger sequencing is highly accurate. The quality of its base calling is typically accessed by using the Phred score, $Q$, calculated using the following equation (*Ewing and Green, 1998*; *Ewing et al., 1998*).

$$Q = -10 \log_{10} E,$$

where $E$ represents the probability of the base-calling error. Phred scores are used to characterize the quality of DNA sequences and filter out low-quality data to ensure the reliability of sequencing results.

Recently, long-read sequencing has emerged as an alternative method to Sanger sequencing for verifying plasmid sequences. Notable among these methods is nanopore sequencing, which uses nanometer-sized holes in polymer membranes (*Wang et al., 2021*; *Agah et al., 2016*). When an electric current is applied across the pore, the passage of DNA molecules through the pore decreases the flux of ions to produce an electrical signal. These changes can be used to determine the sequence of the DNA molecule. Nanopore sequencing offers several advantages over other sequencing technologies, including the ability to sequence long reads, the ability to detect modifications to the DNA molecule, and the potential for real-time sequencing. Another advantage of nanopore sequencing is that it can reduce costs through barcode introduction during library preparation, enabling multiplexing (*Philpott et al., 2021*; *Bayliss et al., 2017*; *Whitford et al., 2022*). Barcoding can be introduced into sheared DNA either by PCR or PCR-free methods such as blunt/TA ligation or barcoded transposase complexes. The barcoded libraries are pooled, and the DNA is delivered to the nanopores on the flow cell by attaching sequencing adaptors containing motor proteins.

A disadvantage of nanopore sequencing is its relatively high error rate, which arises because the signal is sensitive to factors including sample quality and the speed of DNA movement through the pore (*Delahaye and Nicolas, 2021*; *Sahlin, 2021*). In addition, changes in current are elicited not by movement of single bases through the pore but rather by five-nucleotide 'words' known as *k*-mers (*Jain et al., 2015*). Therefore, changes in current do not correspond directly with bases, making base calling and error correction difficult. To compensate for its high error rate, consensus sequencing is typically employed, wherein multiple reads from different pores are aligned to generate a high-quality sequence.

In the Sanger sequencing era, and even still today with the advent of nanopore sequencing, it is common practice not to sequence entire plasmids but only the insert regions. This shortcut persists because insert regions are more prone to mutation than plasmid backbones due to the nature of their generation. Nevertheless, the probability of mutagenesis in the vector backbone is nonzero, and some plasmids can form dimers or multimers (*Currin et al., 2019*). In addition, as plasmid construction efforts have become increasingly complex in recent years, nanopore sequencing is desirable when inserts are long or contain repetitive sequences. Thus, it is most rigorous to sequence entire plasmids, which is impractical using Sanger sequencing but feasible using nanopore sequencing, rather than simply sequencing inserts. However, in most cases Sanger sequencing is still chosen due to its lower cost. For example, Sanger sequencing currently costs ~$4–5 (USD) per reaction, which produces ~900–1000 bases of high-quality data. In contrast, nanopore sequencing currently costs ~$15 (USD) per plasmid through commercial services like Plasmidsaurus, which uses V14 chemistry on PromethION with an R10.4.1 flow cell, typically returning ~100–1000 reads. Therefore, although nanopore sequencing is a powerful and cutting-edge technology, it has not yet advanced to the point of replacing Sanger sequencing.

Theoretically, mixing samples and submitting them in one tube would greatly reduce costs, but such mixing also reduces the quality of the analysis because of two reasons: (1) it is unknown from which sample each nanopore read is derived, and (2) the number of reads per sample will diminish. Typically, in large-scale sequencing such as whole-genome sequencing performed by laboratories that own sequencers, barcoding is used to determine the origin of each read. In addition, each read is aligned against a reference sequence to increase the quality of the analysis. In contrast, in the context of plasmid construction, most users outsource sequencing to core facilities or companies. In this case, it is the third party that performs library preparation for nanopore sequencing; therefore, users typically cannot include barcodes. Finally, whereas reference sequences are not updated frequently in genome sequencing, a new reference plasmid sequence must be used for each analysis in plasmid construction, making it difficult to take advantage of existing platforms for pooling and multiplexing samples.

Here, we have developed a barcode-free, easy-to-use computational approach termed Simple Algorithm for Very Efficient Multiplexing of Oxford Nanopore Experiments for You (SAVEMONEY) that guides users to pool samples for nanopore sequencing and effectively reduces sequencing costs to as low as $2.50 (USD) per plasmid, which is about twice less expensive than one reaction of Sanger sequencing. Our approach involves submitting samples with multiple different plasmids mixed in a single tube and deconvolving the obtained sequencing results while maintaining the quality of the analysis. Instead of using additional barcodes, SAVEMONEY leverages plasmid blueprints (maps), which are in most cases already made by researchers prior to plasmid construction. We found that reads from plasmids that differ by as little as two bases in their sequences can be accurately deconvolved without any additional barcoding. Further, we used Bayesian analysis to increase the quality of analysis from a lower number of reads per sample, adopting strategies commonly used in the analysis of single nucleotide polymorphisms (SNPs) (*Li et al., 2009*). To make our method widely available, we implemented SAVEMONEY on Google Colab and have made code available on GitHub and PyPI. Thus, SAVEMONEY is a straightforward and robust approach for experimental multiplexing and computational de-multiplexing of nanopore plasmid sequencing to accelerate democratization of this powerful technology.

## Results

### Overview of the algorithm

In most cases of plasmid construction, the correct (i.e. expected) sequence is known because the process starts with making a blueprint or a map of the plasmid. In addition, typically only plasmids confirmed to have been largely constructed correctly by rapid and inexpensive restriction enzyme digestion or similar laboratory assays are submitted for sequencing analysis. These pieces of a priori information – 'knowledge of correct, expected sequence of the plasmid' and 'certainty that construction largely proceeded correctly' – can be used to classify the mixed sequence reads and to improve the quality of base calling. Therefore, it should be possible to attain comparable accuracy using a smaller number of reads than that provided in a typical commercial nanopore sequencing sample, enabling barcode-free multiplexing.

The outline of our procedure is as follows: (1) pre-survey, (2) sample submission, and (3) post-analysis (*Figure 1*). The pre-survey step determines the optimal combination of plasmids suitable for mixing prior to submitting for nanopore sequencing. If the sequences of two plasmids are similar, it becomes more difficult to classify reads from each plasmid a posteriori even with the presence of prior information. Therefore, these kinds of plasmids should be submitted as separate samples. For example, multiple colonies from the same plasmid construction effort cannot be submitted together, because they are expected to have the same sequence (i.e. they share an identical blueprint/map). Therefore, our pre-survey algorithm examines the blueprints of the plasmids and generates recommended groupings as outputs, so that similar ones do not fall into the same group.

One key variable that our pre-survey does not explicitly determine is the maximum number of plasmids that may be safely grouped together. Because read quality, assessed by examining quality score distributions, is typically stable, the main variable affecting multiplexing ability is the overall coverage per plasmid. Having too many plasmids in one group will result in too few reads per plasmid and lower quality scores for the final consensus sequence, but this shortcoming can be compensated if the nanopore flow cell produces higher coverage. However, the coverage provided by commercial sequencing services such as Plasmidsaurus varies from the order of $10^2$–$10^3$ and is difficult to predict because each nanopore flow cell has different properties. Therefore, it is not possible to determine the maximum number of plasmids that can be mixed in the pre-survey process. Instead, we project that the theoretical minimum number of reads that is required for the reliable consensus calculation is 30 reads per plasmid (discussed in detail later in 'Maximum number of plasmids that can be mixed' section). In practice, we typically obtain high-quality results by mixing up to six plasmids when submitting samples for sequencing at the Plasmidsaurus service certified by Oxford Nanopore Technologies, which currently typically provides ~200 or more reads per sample, with read length and quality score distribution properties shown in later figures.

Based on the grouping determined in the pre-survey step, the plasmids classified into the same group are then mixed at equal concentrations. If plasmid concentrations are not equal, the coverage

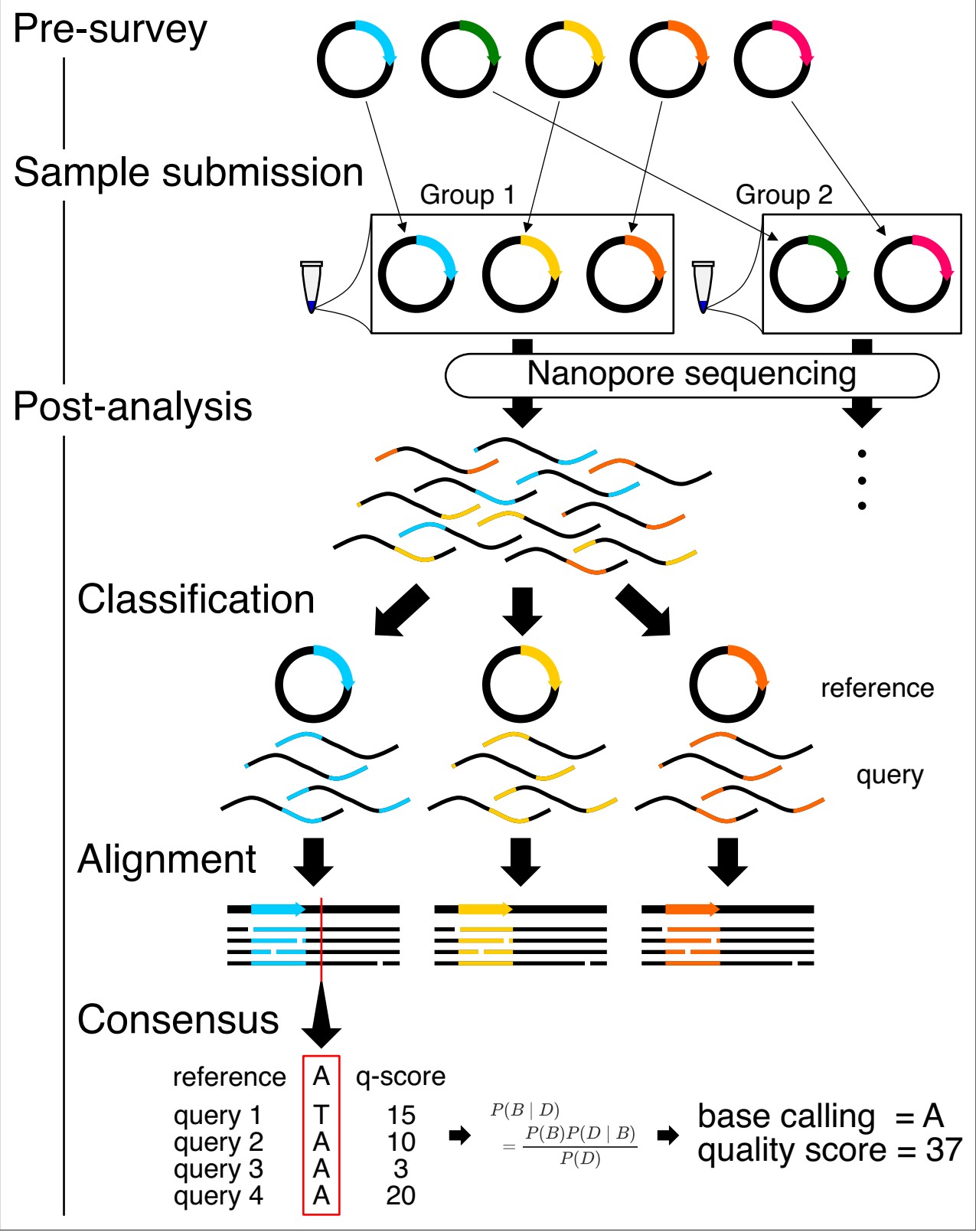

**Figure 1.** Workflow for Simple Algorithm for Very Efficient Multiplexing of Oxford Nanopore Experiments for You (SAVEMONEY). The algorithm consists of three steps: pre-survey, sample submission, and post-analysis. The pre-survey step identifies the optimal combination of plasmids that will permit suitable accuracy for the classification step of the post-analysis. Plasmids with divergent sequences are grouped together, and those with very similar sequences are classified into different groups. After sample submission and sequencing, the post-analysis component, which consists of three different

*Figure 1 continued on next page*

*Figure 1 continued*

steps, is performed to deconvolve the obtained results. Reads (query sequences) are first classified based on their similarity to the plasmid blueprint/map (reference sequence). Reads are then aligned against reference sequences. Finally, consensus sequences and quality scores are calculated based on base calls, quality scores from each read, and the reference sequence, using Bayesian analysis.

of the plasmids with lower concentrations decreases, affecting the reliability of the results. Following mixing of plasmids according to the grouping, samples are submitted for nanopore sequencing according to specific vendor instructions.

After obtaining results from each nanopore sequencing run, deconvolution is then performed as a post-analysis to obtain consensus sequences for each plasmid. The post-analysis uses the following files as inputs: blueprints of the mixed plasmids (reference sequences) and an output FASTQ file of nanopore sequencing results containing base calling and quality scores of each read (query sequences). The algorithm is divided into three steps: (1) classification of reads, (2) alignment of the classified reads, and (3) calculation of the consensus sequence and quality score (*Figure 1*). In each step, reference sequences are used as prior information to increase accuracy and quality. The outputs of the post-analysis are FASTQ files containing consensus sequences and quality scores. Two types of FASTQ files are produced: FASTQ files that use arbitrarily set prior probability of error during the plasmid construction at the last step of the post-analysis and those that do not. Results without prior probability are calculated based on an unbiased analysis, whereas results with prior probability are statistically biased toward the blueprint of the plasmid map based on the prior probability (i.e. an arbitrarily set error rate during PCR, ligation, or assembly). The latter option can be considered as analogous to the case where the peak shape of a base within a Sanger sequencing chromatogram is not clear enough for the automatic base calling and is subsequently (and typically manually) compared to the blueprint of the plasmid to determine the identity of the base. Apart from two FASTQ files, a summary GIF file is also provided, showing the composition of reads matched, mismatched, or determined to be deletions or insertions at each position of the plasmid. Because post-analysis is designed assuming that there are no significant differences (insertions or deletions) between blueprints and the actual samples, it is recommended to always check this summary GIF file to confirm whether the sample meets that assumption.

The pre-survey and the post-analysis algorithms are available via a ready-to-execute Jupyter Notebook on Google Colab or locally executable scripts on GitHub (*Uematsu, 2024*). We have also packaged the script and released it on PyPI to facilitate easy installation and integration by other developers. In the Google Colab version, a function is also included to visualize the classified query sequence along with the reference sequence and the consensus sequence. In addition to analysis of plasmids, SAVEMONEY can also be used for linear DNA such as PCR amplicons and long synthetic DNAs by specifying the 'topology_of_dna' option, though we mainly discuss its use for plasmid sequencing in this paper.

## The pre-survey algorithm

The pre-survey algorithm determines the combination of plasmids that are appropriate for mixing. Its initial step is a series of pairwise alignments of each plasmid. Although in principle any alignment algorithm can be used, we first chose to detect chunks of sequences that are sufficiently long and identical to save computational resources, and we then performed an alignment against sequences between chunks by the classical Smith-Waterman algorithm to ensure the accuracy by dynamic programming (*Smith and Waterman, 1981*). The following parameters were used:

$$\text{match\_score} = 1,$$
$$\text{mismatch\_score} = -2,$$
$$\text{open\_gap\_penalty} = 3,$$
$$\text{extend\_gap\_penalty} = 1.$$

Next, the distance between each plasmid pair was calculated as Levenshtein distance, the minimum number of bases that have to be substituted, deleted, or inserted to change one sequence into the other. Using this framework, the alignment of a plasmid to itself will yield a distance of 0. Calculation of the distances between all plasmid pairs yielded a distance matrix of plasmids. Plasmids with

smaller distances were then classified into the same cluster, which we note is different from the final output of the grouping. Hierarchical clustering was adopted using the distance matrix obtained in the previous step, with the nearest point algorithm chosen as the method for calculating the distance between newly formed clusters (*Figure 2*). Clusters were defined according to a user-defined parameter, *distance_threshold*, which represents the minimum number of bases that must differ between plasmids in the same group. Therefore, all plasmid pairs with distances lower than *distance_threshold* are classified into the same cluster. In practice, lower *distance_threshold* values result in allowing plasmids that are more similar to one another to be classified into different clusters, producing fewer total groups of plasmids for sample submission and thus lowering sequencing costs at the expense of a higher risk of errors during the classification step of the post-analysis algorithm.

Finally, plasmids were classified into groups for sample submission such that plasmids from the same cluster do not fall into the same group and that the minimum distance between plasmids within each group is maximized. To obtain optimal results, we reduced this classification problem to a zero-one integer linear programming problem. First, a rank-3 binary tensor $T \in \{0,1\}^{g \times c \times p}$ was prepared, where the sizes of its dimensions correspond to the number of groups, clusters, and plasmids, and each element takes a value of either 0 or 1. The following constraints were applied to this tensor:

$$
T_{gcp} = \begin{cases} 1 & \text{if } P_p \in C_c, \\ 0 & \text{if } P_p \notin C_c, \end{cases}
$$

$$
\sum_{c,p} T_{gcp} = G_g,
$$

$$
\sum_{p} T_{gcp} \leq 1,
$$

where $P_p$ and $C_c$ indicate plasmid and cluster with the index of $p$ and $c$, respectively, and $G_g$ represents the number of plasmids in the group with the index of $g$, which will be described in more detail below. The constraints represent the following: each plasmid exists only once and belongs to a designated cluster (first constraint), each group contains a designated number of plasmids (second constraint), and plasmids from the same cluster cannot belong to the same group (third constraint). Regarding the first constraint, the cluster to which a plasmid belongs is determined by the results of hierarchical clustering and the values of *distance_threshold* and *number_of_groups*. For the second constraint, the number of plasmids in each group can be set arbitrarily, as long as $\sum_g G_g$ matches the total number of plasmids. However, in the implementation of this algorithm, the number of plasmids in each group was automatically determined to be as equal as possible. In addition to these three constraints, constraints of plasmid pairs that belong to different clusters but should not be distributed into the same group, as expressed by $T_{gc_1p_1} + T_{gc_2p_2} \leq 1$, are sequentially added to a 'forbidden list', where $c_1$ and $c_2$ represent the cluster indices to which $P_{p_1}$ and $P_{p_2}$ belong, respectively. In the algorithm implementation, in addition to the 'forbidden list', an 'exclusion list' was also used to prevent plasmid pairs from being added to the 'forbidden list'. Until all plasmid pairs that belong to different clusters were included in one of the two lists, the following steps were repeated:

1. Among the plasmid pairs belonging to different clusters, the pair with the shortest distance that is not included in either the 'forbidden list' or the 'exclusion list' is added to the 'forbidden list'.
2. Find a feasible solution for $T \in \{0,1\}^{g \times c \times p}$ under the given constraints. If a solution is found, go back to step 1 and repeat the process. If no solution is found, remove the last plasmid pairs from the 'forbidden list', and instead add it to the 'exclusion list'.

This process ultimately leads to the final grouping result.

Examples of the pre-survey results against 14 plasmids are displayed in *Figure 2*, together with the dendrogram used during the step of hierarchical clustering. When a *distance_threshold* value of 5 is applied without specifying *number_of_groups*, plasmids were classified into groups of two, because there are two clusters (clusters 3 and 12 in *Figure 2a*) containing two plasmids, and plasmids within each cluster have to be distributed into different groups. The minimum distance within each group was 6 (P8 and P10 in Group 2 in *Figure 2a*) under this condition, but the value increased to 10 (P1 and P4 in Group 1 in *Figure 2b*) by specifying *number_of_groups* as 3, indicating that grouping is more

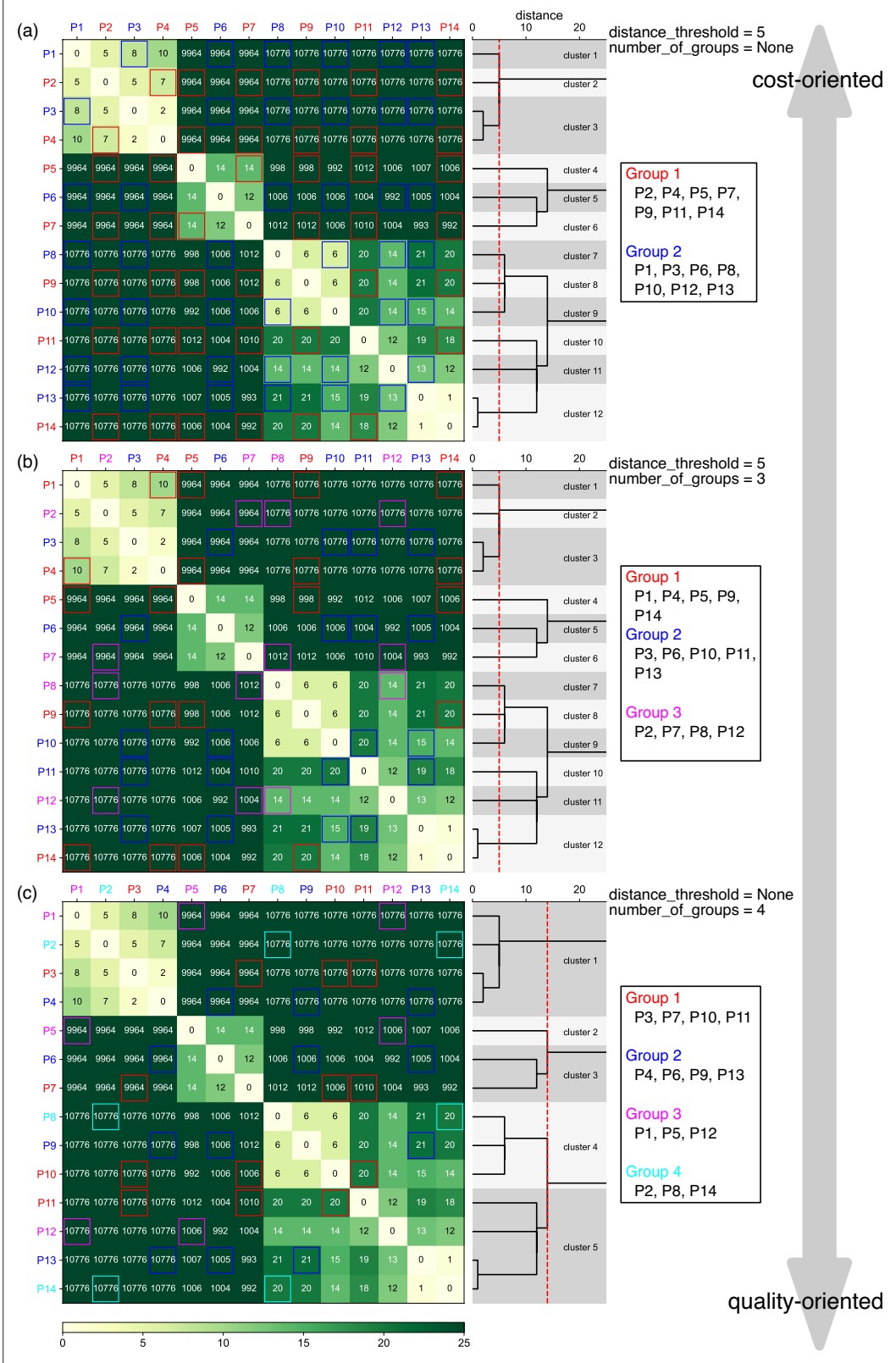

**Figure 2.** Examples of the pre-survey outputs. Sequences of 14 different plasmids were analyzed with the indicated *distance_threshold* and *number_of_groups* values. Levenshtein distance between each plasmid pair is displayed in heatmaps, which were subsequently used to generate the dendrograms displayed on the right side of the heatmap. The dotted red lines in the dendrogram represent the *distance_threshold* values, enabling

*Figure 2 continued on next page*

*Figure 2 continued*

visualization of the results of clustering, i.e., plasmids with distances less than the red lines were classified in the same cluster. Based on this clustering results of similar plasmids, plasmids were classified into groups for sequencing submission, with groups displayed in red and blue (**a**), red, blue, and magenta (**b**), or red, blue, magenta, and cyan (**c**). Levenshtein distance between each plasmid is also emphasized by the colored frames within each group, and plasmids classified into the same cluster are grouped in different groups. Note that P1–P14 here are different from example plasmids used in *Figures 3 and 4*.

The online version of this article includes the following source data for figure 2:

**Source data 1.** Results of pre-survey used to make *Figure 2a*.

**Source data 2.** Results of pre-survey used to make *Figure 2b*.

**Source data 3.** Results of pre-survey used to make *Figure 2c*.

quality oriented. By contrast, users can determine the grouping by only specifying *number_of_groups* (*Figure 2c*). When the value was set to 4, the algorithm automatically determines the minimum *distance_threshold*, which in this case was 14, to satisfy the provided *number_of_groups*. The minimum distance within each group was 20 (P10 and P11 in Group 1 in *Figure 2c*), indicating the grouping was of the highest quality among the three per-surveys. These results show that lower *distance_threshold* or *number_of_groups* values lead to reduced sequencing costs. Conversely, plasmids with higher degrees of similarity will be incorporated into the same group under such settings, which may slightly reduce the reliability of the post-analysis. However, we found that *distance_threshold* values can be set down to 2 based on the assessment of the limitation of post-analysis, which will be discussed later (see 'Maximum similarity allowed for mixing' section for details).

## The post-analysis algorithm

To classify reads from each pore, the alignment was first performed against the reference plasmid $p_i$ and the query sequence $q_k$ from each pore using the same parameters as described in the pre-survey algorithm. Then, the normalized alignment score, $a_{p_i,q_k}$, was calculated by dividing the alignment score by the length of $p_i$. The plasmid to which the query is assigned, $p_{q_k}$, was determined as follows:

$$p_{q_k} = \underset{p_i \in P}{\mathrm{argmin}}\, a_{p_i,q_k},$$

where $P$ indicates the set of reference plasmids that were mixed. However, if $a_{p_{q_k},q_k}$ was lower than *score_threshold*, a user-defined value that represents a cutoff for short reads (see below), the read $q_k$ was excluded to increase the quality of subsequent post-analysis. Furthermore, the read $q_k$ showing the same normalized alignment score against more than one plasmid was also omitted, because such a read does not contain enough information to determine the plasmid from which it originated. Lastly, the read $q_k$ more than two times longer than the length of the assigned reference plasmid $p_{q_k}$, and that showed a higher normalized alignment score than 1 was excluded to omit the read from plasmid multimer.

Four examples of the results from four different groups of plasmids are displayed in *Figure 3* (Group I) and *Figure 4* (Groups II, III, and IV). Group I involves multiplex sequencing of six plasmids with only modest similarity, and Groups II–IV contain two plasmids each with high similarity. (Note that for *Figures 2–4*, we have elected to name the plasmids similarly [e.g. P1, P2, etc.] for simplicity; however, they are all distinct, i.e., P1 from *Figure 2* is different from P1 in *Figure 3*.) The pre-survey results for Group I are shown in *Figure 3a*, indicating two clusters of four (P1–P4) and two (P5–P6) similar plasmids. In reality, each cluster contains plasmids that share a common vector, illustrating how plasmids sharing common vector backbones but different inserts are moderately similar but still suitable for multiplexing, as their distances are far greater than 20, which is a very safe, quality-oriented cutoff value shown in *Figure 2c*. These six plasmids were mixed and analyzed as a single sample by nanopore sequencing. *Figure 3b* represents the general quality check of the results for each plasmid: distributions of read length and quality scores.

Reads assigned to each reference plasmid showed sharp read length distributions, indicating the accurate classification of the reads. In addition, the quality score distributions of reads assigned to each reference plasmid are very similar, indicating that each plasmid was sequenced with almost the same

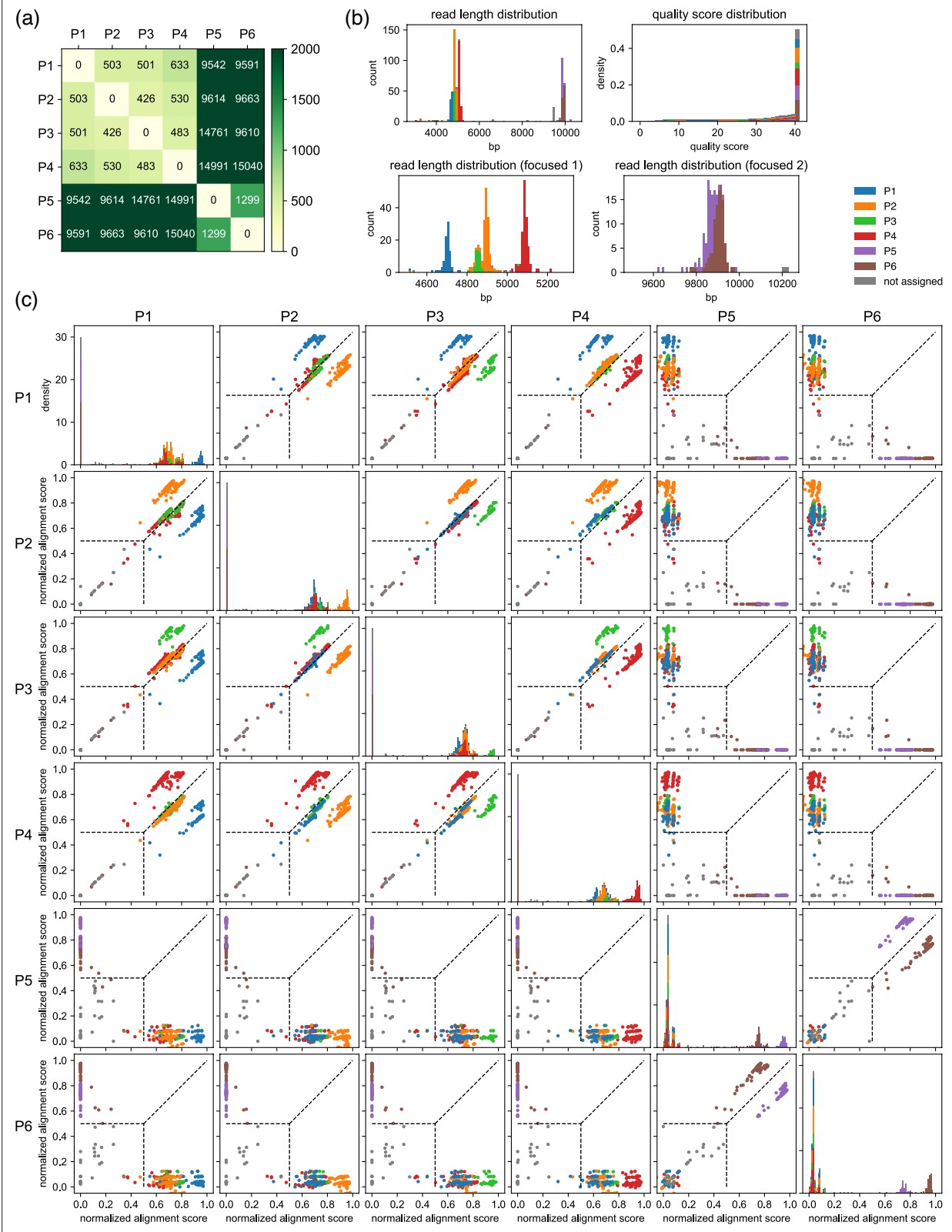

**Figure 3.** Example results after the classification step for a set of six moderately related plasmids. (**a**) Results of the pre-survey against plasmid sets used in (b, c). (**b**) Read length and the quality score distributions. (**c**) Scatter plots of normalized alignment scores. Normalized alignment scores were calculated for each read over all reference plasmids and displayed as scatter plots. Density plots of the normalized alignment scores are also displayed for each reference plasmid in the diagonal panel, which is the projection of each scatter plots against horizontal axes. The *y*-axis ranges of these

*Figure 3 continued on next page*

*Figure 3 continued*

diagonal panels are shared. The vertical and the horizontal positions of dashed lines correspond to *score_threshold* value. These data depict results of classification performed with a *score_threshold* value of 0.5. Note that P1–P6 here are different from example plasmids used in *Figures 2 and 4*.

The online version of this article includes the following source data for figure 3:

**Source data 1.** Fastq file containing original data used to make *Figure 3b and c*.

**Source data 2.** Csv file containing raw data used to make *Figure 3b*.

**Source data 3.** Csv file containing raw data used to make *Figure 3c*.

quality. Note, however, that the number of reads was not identical for each plasmid, which is reflected by the total histogram area of each colored portion in the read length distribution graphs. Scatter plots of normalized alignment score, $a_{p_i,q_k}$, for each reference plasmid pair are shown in *Figure 3c*, which is useful to adjust *score_threshold*. From the graphs, it is apparent that if *score_threshold* is set higher, the classification accuracy would increase, at the expense of lowering the number of reads assigned to each reference plasmid. For typical experiments, we have found 0.5 to be a reasonable value for *score_threshold*, but users can fine-tune this parameter to make the most out of the acquired data. For example, if the total number of reads was small but each plasmid was sufficiently different, the threshold can be lowered to increase the number of reads that are passed on to the next step of analysis. Conversely, if the total number of reads is large and the plasmids are highly similar with each other, the quality of subsequent analysis can be improved by raising the threshold. *Figure 3c* shows the well-separated scatter plots of each read against each reference plasmid, indicating successful classification of reads. In this experiment, a sufficient number of reads (90, 154, 57, 165, 73, 107, and 50 each for P1, P2, P3, P4, P5, P6, and unassigned, respectively) were obtained for each plasmid, which ensures reliability in the subsequent calculation of the consensus sequence.

The results of different combinations of plasmids with high sequence similarity are displayed in *Figure 4*. The biggest concern in this case is whether each read is accurately classified to each reference plasmid. Assuming the case where two plasmids are mixed, theoretically, the classification of a read to either plasmid is determined solely by the regions where the plasmids differ in sequence. This result occurs because errors in regions where the plasmids have identical sequences have the same effects on the normalized alignment score for both plasmids, i.e., errors in such regions decrease the normalized alignment score in the same manner for both plasmids. Hence, the accurate classification can be achieved for plasmids that differ by even a single base, considering the high accuracy of recent nanopore sequencing. To demonstrate this outcome, we mixed and submitted three sets of two plasmids (set 1: P1–P2, set 2: P3–P4, and set 3: P5–P6) with Levenshtein distances of only 1, 2, and 3 as a single sample (*Figure 4a–c*). The corresponding results after the classification step of the post-analysis are displayed in *Figure 4d–f*. Because the plasmids are almost identical, most of the data points are nearly overlapping the *y=x* line in the scatter plot of the normalized alignment score (*Figure 4d–f*, top). However, a magnified view clearly shows the deviation of data points from *y=x* line, indicating that reliable separation of many reads can be achieved (*Figure 4d–f*, bottom).

Further, we performed a quantitative analysis to estimate the rate of incorrect classification. First, the reads covering the regions where the plasmids differ in sequence were extracted. The numbers of such reads were 239 out of 303, 262 out of 297, and 2212 out of 2359 reads for sets 1, 2, and 3, respectively, and the breakdowns of classification results are shown in *Figure 4g–i*. The histograms on top also show the number of reads with a normalized alignment score below *score_threshold* (0.5 in this case) in light gray, but the numbers are all very small, suggesting high sequencing quality. The rotated heatmaps below provide more detailed breakdowns of the reads represented in the histograms above. Using these structured data in the rotated heatmaps, we performed a simple fitting using the least squares method, with the number of reads originating from each plasmid and the nanopore error rate as variables. For simplicity, we approximate that the probability of a nanopore-based base calling error is the same independent of the identity of the base (e.g. when errors occur for true base A, the probability of base calling being T, G, or C is one over 3 for each) and that there are no deletion or insertion errors in base calling. The summary of the fitting results is shown in *Figure 4j–l*. The upper and lower rotated heatmaps in *Figure 4j–l* represent the breakdown of reads originating from the plasmid with the lower and higher numbering, respectively. These fitting results allowed us to calculate the ratio of reads classified under a reference plasmid that differs from their

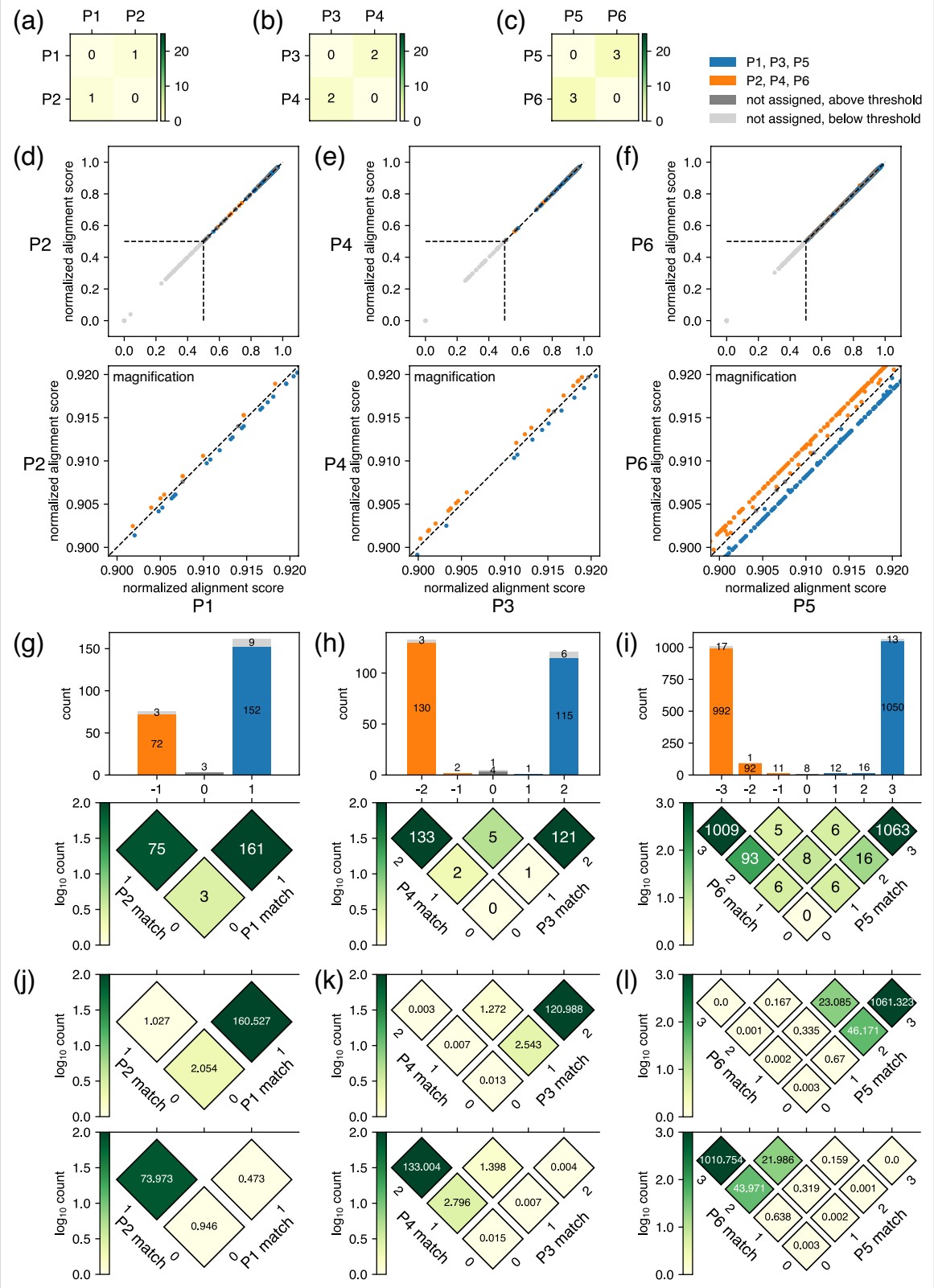

**Figure 4.** Example results after the classification step for closely related plasmids. (**a–c**) Results of the pre-survey against plasmid sets. (**d–f**) Scatter plots of normalized alignment scores for the plasmid pairs. The vertical and the horizontal positions of dashed lines correspond to *score_threshold* values. These data depict results of classification performed with a *score_threshold* value of 0.5. (**g–i**) Breakdowns of reads covering the regions where the plasmids differ in sequence. In the rotated heatmap at the bottom of (**g**), the axis labeled as 'P1 match' represents the number of bases matching the P1

*Figure 4 continued on next page*

*Figure 4 continued*

sequence in the regions where the sequences of P1 and P2 differ, whereas the axis labeled as 'P2 match' represents the equivalent for P2. The values in each cell represent the number of observed reads matching the values of the two axes at that position. The subtraction of 'P1 match' from 'P2 match' is represented by the horizontal axis, which is also shared with the *x*-axis of the histogram on top, where the sum projection of the heatmap is displayed. In the histogram, the breakdown of classification is represented by color: blue for reads classified to P1, orange for reads classified to P2, gray for unclassified reads with a score above *score_threshold*, and light gray for unclassified reads with a score below *score_threshold*. Because the normalized alignment scores for the two plasmids are the same where the value on the horizontal axis is 0, reads are not classified to either plasmid; therefore, the middle bar of the histogram is colored with either gray or light gray. The same interpretation applies to (**h**) and (**i**). (**j–l**) Summary of the fitting results. Based on the estimated parameters displayed in *Table 1*, the rotated heatmaps representing the breakdowns of reads originating from P1, P3, and P5 (upper panels) and P2, P4, and P6 (lower panels) were generated. Note that P1–P6 here are different from example plasmids used in *Figures 2 and 3*.

The online version of this article includes the following source data for figure 4:

**Source data 1.** Fastq file containing original data used to make *Figure 4d and g*.

**Source data 2.** Fastq file containing original data used to make *Figure 4e and h*.

**Source data 3.** Fastq file containing original data used to make *Figure 4f and i*.

**Source data 4.** Csv file containing raw data used to make *Figure 4d and g*.

**Source data 5.** Csv file containing raw data used to make *Figure 4e and h*.

**Source data 6.** Csv file containing raw data used to make *Figure 4f and i*.

actual origin, which was represented as 'incorrect assignment rate' in *Table 1*. The values indicate that plasmids that differ even by a single base (set 1) could be classified to a sufficient confidence level with around 0.03–1.4% incorrect classification. The ratio further goes down when Levenshtein distance was increased to 2 (set 2) or 3 (set 3), transforming the base calling error ratio of 1.55% and 2.13% into the incorrect classification ratio of 0.0074–0.0089% and 0.0143–0.0158%, respectively. These results indicate that the nanopore base calling error negligibly affects the classification step, especially for plasmid pairs differing by two or more bases. The results also suggest that the error should not affect the subsequent analysis to obtain consensus base calling. However, we recommend mixing plasmids that differ by at least two bases, i.e., the *distance_threshold* value should be at least 2, due to the following reasons. First, although this low percentage of incorrect assignment is unlikely to affect the consensus base calling even with plasmids with 1 base difference, it does affect the consensus quality score, and our algorithm does not make any correction for it. Second, although the average error rate of current nanopore technology is low, it can be increased to up to ~40% for a few specific sequences, such as Dam methylation, and this number could result in incorrect consensus base calling. Third, if

**Table 1.** The fitting results summarized in *Figure 4j–l*.

The values for the fitted parameters are displayed in 'Error rate' and 'Total reads' columns. The rotated heatmaps in *Figure 4j–l* were generated based on these estimated parameters. For P1, P3, and P5, the sum of values in heatmap cells whose location on the horizontal axis are above 0, below 0, and 0 are shown as 'Correctly classified reads', 'Wrongly classified reads', and 'Reads not classified' columns, respectively. For P2, P4, and P6, below 0, above 0, and 0 are shown as 'Correctly classified reads', 'Wrongly classified reads', and 'Reads not classified' columns, respectively. Finally, the values in 'Rate of incorrect classification' columns for each plasmid were calculated by dividing the values of 'Wrongly classified reads' for the other plasmids in the same set by the total number of reads estimated to be classified to the focusing plasmid, which is different from values displayed in the 'Total reads' column. Specific equations are provided in the footnote to the table.

| | | Error rate | Total reads | Correctly classified reads | Wrongly classified reads | Reads not classified | Rate of incorrect classification |
|---|---|---|---|---|---|---|---|
| | P1 | | 163.607 | 160.527[a1] | 1.027[b1] | 2.054 | 0.002939[c1] |
| Set 1 | P2 | 0.0188 | 75.393 | 73.973[a2] | 0.473[b2] | 0.946 | 0.013691[c2] |
| | P3 | | 124.826 | 123.531[a3] | 0.010[b3] | 1.285 | 0.000089[c3] |
| Set 2 | P4 | 0.0155 | 137.223 | 136.000[a4] | 0.011[b4] | 1.413 | 0.000074[c4] |
| | P5 | | 1131.757 | 1131.249[a5] | 0.170[b5] | 0.338 | 0.000143[c5] |
| Set 3 | P6 | 0.0213 | 1077.833 | 1077.349[a6] | 0.162[b6] | 0.322 | 0.000158[c6] |

c1=b2/(a1+b2); c2=b1/(a2+b1); c3=b4/(a3+b4); c4=b3/(a4+b3); c5=b6/(a5+b6); c6=b5/(a6+b5).

only one base differs, that base is the only piece of information that can correctly classify reads, indicating that an unexpected mutation in that position during the plasmid construction would ruin the classification. Therefore, we set the minimum *distance_threshold* value to 2 in the pre-survey, and we recommend setting it to a higher value when possible.

After the classification step, each read is aligned against the corresponding reference sequence, and then a final post-analysis step is executed to obtain the consensus sequence and quality score. Here, the aligned query sequences, quality scores of each read, and prior information are combined using Bayesian analysis, similar to previously reported methods to detect SNPs (*Li et al., 2009*). When generating consensus sequences in SAVEMONEY, two types of prior information are used: (1) the error rate during plasmid construction (i.e. an arbitrarily set error rate during PCR, ligation, or assembly), and (2) the characteristics of the nanopore reads, i.e., error rate and distribution of quality score for each base. For example, assume that the correct base at a specific position in the blueprint of the plasmid is A, and that 10 reads were obtained for the corresponding part, of which eight had base calls of A and the other two had G. In this case, the following cases can be considered:

1. True base is A, and error happened for 2 reads returning G.
2. True base is T, and error happened for all 10 reads.
3. True base is C, and error happened for all 10 reads.
4. True base is G, and error happened for 8 reads returning A.

By using prior information, each of the above probabilities can be calculated. It is reasonable to adopt the case with the highest probability among them and use it as the consensus base call. Note that the possibility of deletion and insertion is not considered in the above example to make the example simple, but they are implemented in the actual script.

Although quality scores of each read from Oxford Nanopore Technologies do not exactly follow Phred scores, they are approaching Phred scores in recent years (*Delahaye and Nicolas, 2021*; *Laver et al., 2015*). Therefore, for practical purposes, we elected to consider the quality scores as Phred scores, while accepting a small error in the calculation of consensus quality score to some extent. Based on this assumption, the error rate, $E_{basecall=B}$, when the consensus base calling is $B$ at a specific location, can be calculated as follows using Bayes' theorem:

$$
\begin{aligned}
E_{basecall=B} &= 1 - P\left(B \mid D_1, D_2, \cdots\right) \\
&= 1 - \frac{P\left(B\right) P\left(D_1, D_2, \cdots \mid B\right)}{P\left(D_1, D_2, \cdots\right)},
\end{aligned}
$$

where $P\left(B\right)$ represents the probability that the true base is $B$ at the specific location of the plasmid, and $D_k$ represents the data obtained from pore $k$. The prior probability $P\left(B\right)$ can be arbitrarily set from the error ratio during plasmid construction, which corresponds to the first piece of prior information. Here, the likelihood $P\left(D_1, D_2, \cdots \mid B\right)$ and the probability of obtaining the data $P\left(D_1, D_2, \cdots\right)$ can be calculated as follows:

$$
\begin{aligned}
P\left(D_1, D_2, \cdots \mid B\right) &= \prod_k P\left(D_k \mid B\right), \\
P\left(D_1, D_2, \cdots\right) &= \sum_{B'} P\left(B', D_1, D_2, \cdots\right) \\
&= \sum_{B'} \left[P\left(B'\right) \prod_k P\left(D_k \mid B'\right)\right].
\end{aligned}
$$

This conversion is guaranteed under the condition that the data obtained by each pore is independent if the true base is known. Further, the likelihood $P\left(D_k \mid B\right)$ can be converted as follows:

$$
\begin{aligned}
P\left(D_k \mid B\right) &= P\left(Q_k, b_k \mid B\right) \\
&= P\left(Q_k \mid b_k, B\right) P\left(b_k \mid B\right),
\end{aligned}
$$

where $b_k$ and $Q_k$ represent the base calling and the quality score obtained from pore $k$, respectively. Here, $P\left(Q_k \mid b_k, B\right)$ and $P\left(b_k \mid B\right)$ can be calculated based on the quality score distribution and the error ratio of the nanopore sequencing, respectively, which corresponds to a second piece of prior information. These characteristics can be obtained from nanopore sequencing data of plasmids with

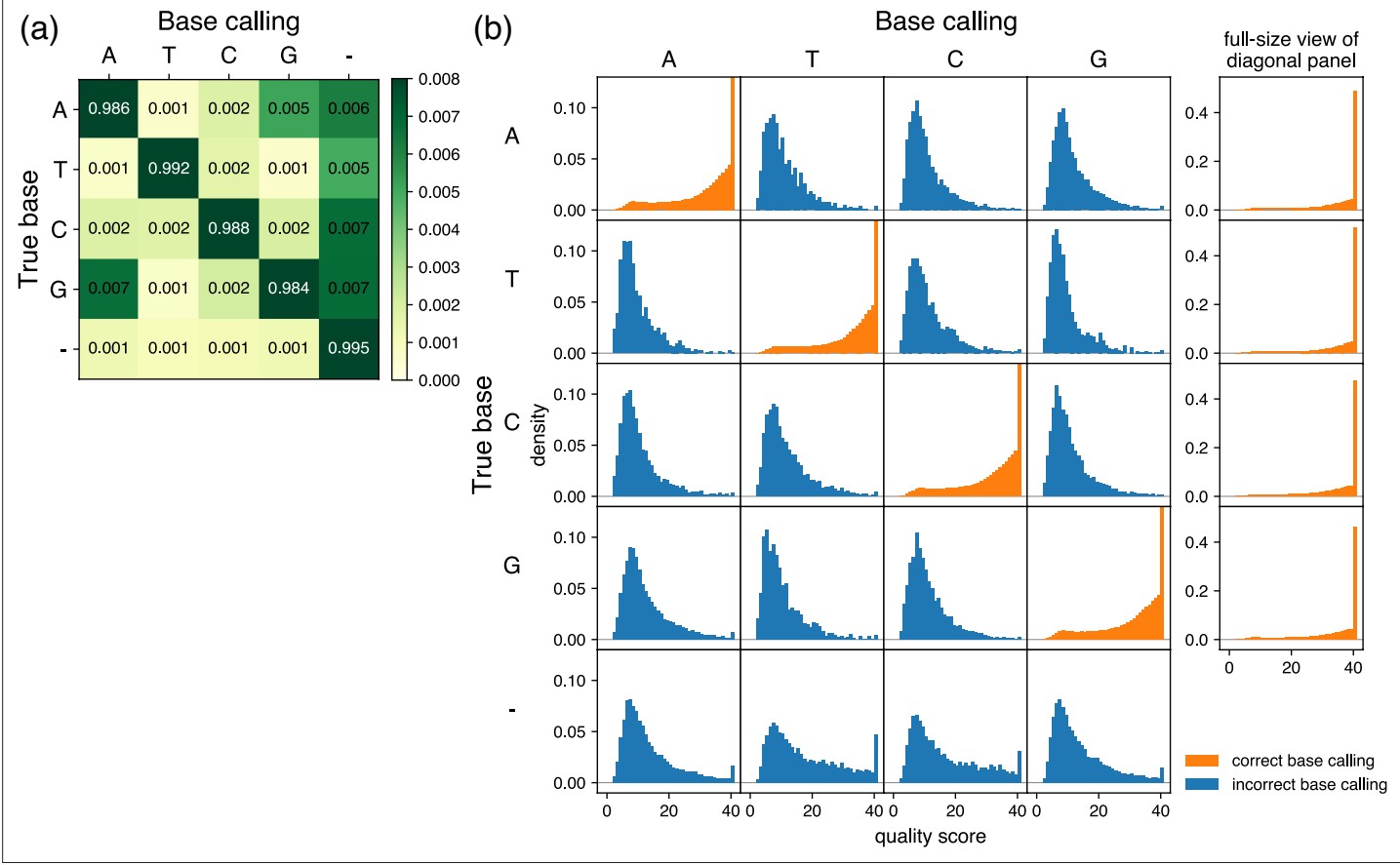

**Figure 5.** Characteristics of base calling used for prior information. (**a**) Grid showing error ratios for each base calling event. Based on the results obtained from samples analyzed by R10.4.1 flow cells with V14 library preparation chemistry by Oxford Nanopore Technologies via the Plasmidsaurus service, the frequency was analyzed for base calling of each pore (column labels) and the results of the consensus sequence (row labels) at each position. In the context of base calling, '–' represents bases that were base-called in the consensus sequence but skipped in the reads from each pore. In the context of consensus sequencing, '–' represents bases that do not appear in the consensus sequence but were base-called from pores. The color of the diagonal panels is saturated because of the contrast range focusing on subtle differences of the non-diagonal panels. Of note, the sum of the rows is 1, but the sum of the displayed numbers may be slightly different from 1 because the fourth decimal place is rounded in the grid. (**b**) Quality score distributions for each base calling event. The base calling of each pore (column labels) and the results of the consensus sequence (row labels) at each position were classified, and probability density plots and quality scores were calculated and displayed. The y-axis is shared by all panels and is scaled to focus on panels in which the true base and base calling are not the same (incorrect base calling, blue). Therefore, the density of maximum quality score is out of the range of the display area in the diagonal panels (correct base calling, orange), and full-size plots are provided to the right. Note that there are no density plots when base calling was skipped in the reads from each pore (column corresponding to the label '–' in (**a**)).

The online version of this article includes the following source data for figure 5:

**Source data 1.** Text file containing raw data used to make *Figure 5*.

known sequences, or, in practice, with unknown sequences by considering the consensus sequence as a known sequence. Although these characteristics change depending on the types of flow cell and library preparation chemistry, in this paper we proceed based on the characteristics of R10.4.1 flow cells with V14 library preparation chemistry by Oxford Nanopore Technologies that are currently used for sequencing via the Plasmidsaurus service, but the principle should be the same with other long-read sequencers. The representative quality score distribution and the error ratio we obtained by analyzing one plasmid by nanopore sequencing are displayed in *Figure 5*. Thus, the final consensus base calling, $B_{consensus}$, and the consensus Phred score, $Q_{consensus}$, can be calculated as follows:

$$B_{consensus} = \underset{B}{\arg\min}\, E_{basecall=B},$$

$$Q_{consensus} = -10\log_{10} E_{basecall=B_{consensus}}.$$

## Maximum similarity allowed for mixing

The extent to which similar plasmids can be mixed (i.e. how low the *distance_threshold* value can be in the pre-survey step) is greatly affected by the resolution of the 'classification' in the post-analysis. In this step, the use of prior information is important. Assuming that two plasmids, $p_1$ and $p_2$, with different sequences, are mixed and analyzed by nanopore sequencing. If the data from one read from one pore returns a sequence similar to $p_1$, the following two cases are possible:

1. The DNA that passed through the pore was derived from $p_1$ and returned a sequence similar to $p_1$.
2. The DNA that passed through the pore was derived from $p_2$, but the pore was inaccurate and base calling error occurred frequently. By chance, it returned a sequence that is similar to $p_1$.

Intuitively, case 1 is the correct answer, but a more precise expression is that the probability of case 2 is extremely low if plasmid $p_1$ and $p_2$ are 'sufficiently different', so it is safe to consider only the possibility of case 1 in practice. This example implies that the prior information of known plasmid sequences in samples improves the accuracy of the classification of the reads from each pore. It is difficult to perform such a classification process with versatility and high accuracy in the absence of prior information.

To specifically determine what kinds of plasmid pairs are 'sufficiently different', detailed analysis was performed. First, the percentage of reads with correct base calling was calculated for each position of plasmids and the distribution was obtained (*Figure 6a*). More than 98% of positions exhibited a correct base calling rate over 0.9, but a small number (0.1%) of them showed low correct rates below 0.7. Averaged quality score distributions of high (>0.9) and low (<0.7) correct rates are displayed in *Figure 6b*. In the former case, most reads showed the maximum quality score, whereas in the latter case, the percentage is lower. The rate of correct base calling, incorrect base calling, and base calling of deletion were 67.1%, 19.0%, and 13.9%, respectively. *Figure 6c* shows the bases that were enriched around 5-mers of positions that showed correct rates lower than 0.7 in the 'worst-case scenario' indicated in *Figure 6a*. Using this 'worst-case scenario', the probability of incorrect classification when data from one read is classified to $p_1$ was calculated as follows:

$$P\left(p_2 \mid D = p_1\right) = \frac{P\left(D = p_1 \mid p_2\right) P\left(p_2\right)}{P\left(D = p_1\right)},$$

where $P\left(D = p_1\right)$ represents the probability that one read was classified as $p_1$, i.e., normalized alignment score of the read was higher for $p_1$ than $p_2$. The denominator $P\left(D = p_1\right)$ can be transformed as follows:

$$P\left(D = p_1\right) = \sum_{k=1,2} P\left(p_k\right) P\left(D = p_1 \mid p_k\right).$$

Therefore, the plot of $P\left(p_2 \mid D = p_1\right)$ can be calculated and the results are displayed in *Figure 6d*, which varies the number of bases that are different between the plasmids.

When $p_1$ and $p_2$ differ by only one base, the percentage of incorrect classification was estimated to be 39% in this rarely occurring (0.1% frequency) 'worst-case scenario'. When the sequence differs by two bases, this percentage drops to 22%. This value is sufficiently smaller than 50% and can be considered to have no effect on the consensus sequence. Also, this percentage is calculated assuming that the two bases are both the 'worst-case scenario', i.e., the 0.1% case shown in *Figure 6a*. That means the chance of this happening for two bases that differ in $p_1$ and $p_2$ is $10^{-6}$, which we consider to be small enough to not be a cause for concern. Thus, in most cases, the percentage of incorrect classification will be much lower even if there are only two base differences between $p_1$ and $p_2$. In fact, the percentage of misclassified plasmids was estimated as less than 0.3% at the highest for a two-nucleotide difference from experimental data in *Figure 4e–h*. Therefore, we ascertain that a *distance_threshold* value of 2 or more during the pre-survey is sufficient for reliable post-analysis.

## Maximum number of plasmids that can be mixed

The question of how many plasmids can be safely mixed together can be partially replaced by the question of how many reads are required at minimum to obtain a reliable consensus sequence. To

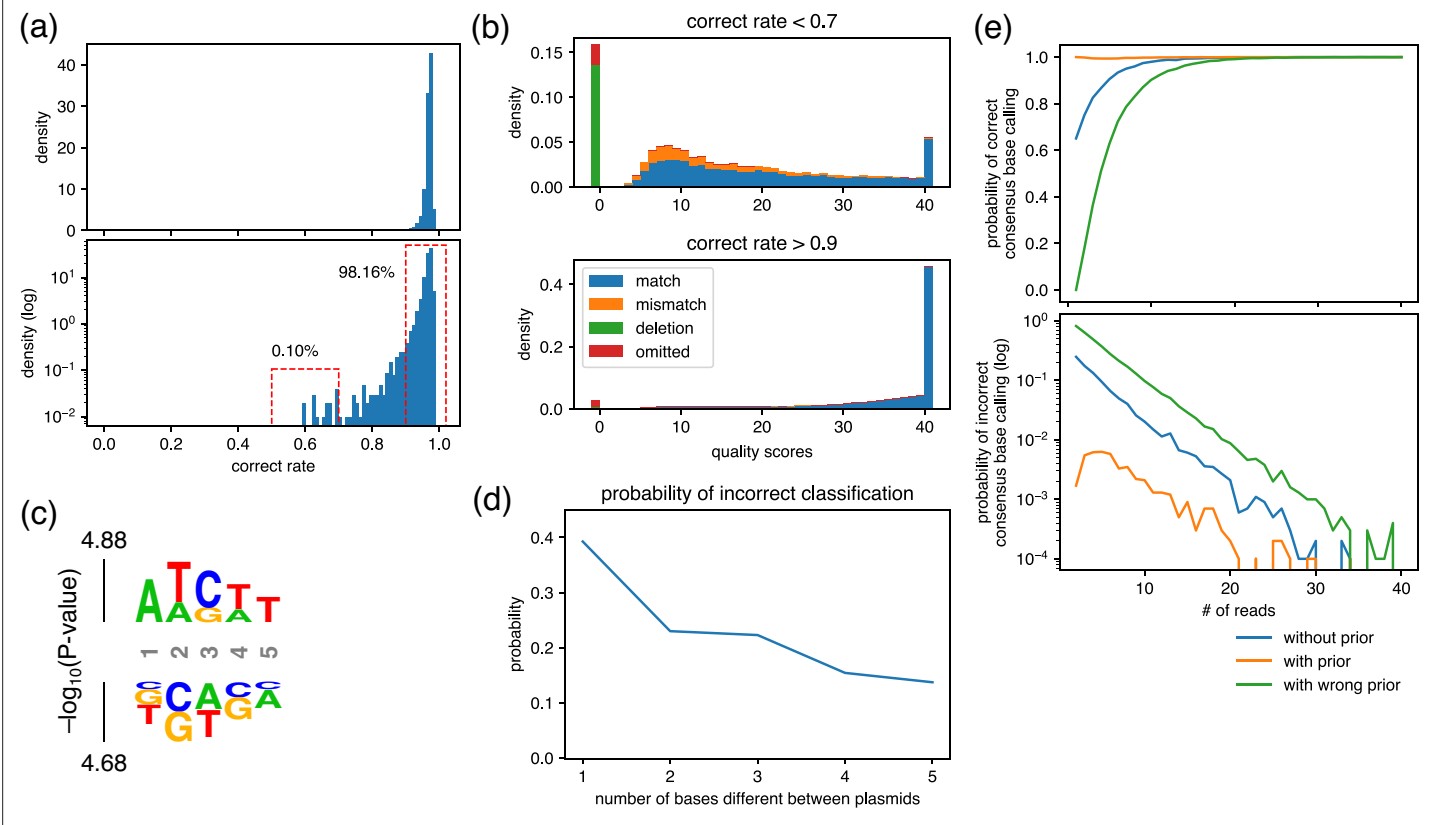

**Figure 6.** Analysis for the maximum similarity between plasmids that can be mixed and the minimum number of required reads. (**a**) Density plot of the rate of reads with correct base calling. The rate was calculated at each position of plasmids and displayed using representative nanopore sequencing results. (**b**) Averaged quality score distribution of reads with correct rate of less than 0.7 (upper panel) and more than 0.9 (bottom panel). The corresponding regions are displayed with dashed red frames in (**a**). Of note, 'omitted' represents reads that did not cover the focused position. (**c**) Probability logo plot. Statistical significance ($-\log_{10}$[p-value]) was calculated for a 5-mer around the positions that showed correct rate lower than 0.7 in (**a**) using those that showed more than 0.9 as a background. Enriched residues are stacked on the top, whereas depleted residues are stacked on the bottom. (**d**) Estimated probability of incorrect classification. Based on the match/mismatch/deletion ratio of reads obtained in the 'worst-case scenario', i.e., top panel in (**b**), the probability of incorrect classification of a read was calculated assuming that two plasmids that differ by the indicated base(s) were mixed. (**e**) Estimated probability of correct/incorrect consensus base calling. Based on the quality score distribution obtained in the 'worst-case scenario', i.e., top panel in (**b**), the indicated number of reads were generated in silico, and the consensus base calling was calculated using Simple Algorithm for Very Efficient Multiplexing of Oxford Nanopore Experiments for You (SAVEMONEY). The simulation was performed 10,000 times for each condition to calculate the probability of correct/incorrect consensus base calling.

The online version of this article includes the following source data for figure 6:

**Source code 1.** Source code used to make *Figure 6d*.

**Source code 2.** Source code used to make *Figure 6e*.

**Source data 1.** Csv file containing raw data used to make *Figure 6a*.

**Source data 2.** Csv file containing raw data used to make *Figure 6b*.

**Source data 3.** Text file containing raw data used to make *Figure 6c*.

**Source data 4.** Csv file containing raw data used to make *Figure 6e*.

answer this question, detailed analysis was performed again using the 'worst-case scenario'. We simulated nanopore base calling according to quality score distribution when the correct rate was less than 0.7 (*Figure 6b*, top panel). For a simulation with 20 reads, as an example, 20 sets of quality scores and read types (match, mismatch, deletion, omitted) were generated according to the distribution, and the consensus base calling and quality scores were calculated using SAVEMONEY. Sampling of 10,000 events was performed in each condition over 1–40 reads to calculate the probability of correct/incorrect consensus base calling (*Figure 6e*). The consensus was calculated in three ways: (1) without prior probability for the unbiased base calling (blue), (2) with prior probability to incorporate an arbitrarily

set plasmid construction error rate (orange), and (3) with wrong prior probability for testing error detection (green). The first two conditions correspond to the two outputs from SAVEMONEY, and the last condition indicates the sensitivity of 'with prior probability' analysis to detect errors when the reference sequence differs from the actual plasmid.

The results show that the ratio of incorrect base calling decreases exponentially (i.e. linearly on the logarithmic plot), and all three lines reach an error rate of less than 0.01 after more than 30 reads. This analysis shows that, even in the 'worst-case scenario', which occurs with a frequency of ~0.1% (*Figure 6a*), the probability of incorrect base calling is less than 0.1%, indicating that the net probability is ~$10^{-6}$. This value is small enough to be negligible even considering the size of usual plasmids (on the order of $10^4$ bases). Therefore, when samples are submitted to a nanopore sequencing service such as Plasmidsaurus, which typically provides a minimum number of ~200 reads per sample, we have found that mixing of up to six plasmids is routinely possible. These conditions bring down the current sequencing cost per plasmid to lower than a single Sanger sequencing run. Further savings and efficiency come from the costs associated with multiple Sanger sequencing runs for inserts longer than 1 kb and re-analysis of occasional failed Sanger sequencing runs. For operators of long-read sequencing, such analysis may enable conservation of expensive reagents. Overall, we expect that the principles underlying our computational approach will accelerate widespread adoption of whole-plasmid sequencing.

## Discussion

In this study, we developed a versatile pipeline that allows end users to easily prepare optimal pooling of plasmids and perform computational de-multiplexing of long-read sequencing results of multi-plexed samples. During plasmid construction, a restriction enzyme digestion test can typically be used to confirm that the plasmid construction was largely successful, but such verification by test digestion is not 100% accurate, necessitating sequencing for ultimate verification of the plasmid sequence. Using Sanger sequencing, Phred scores drop as the read lengths approach ~1000 bases, and accurate base calling becomes difficult without any information. However, in many cases, higher confidence can be obtained by comparing raw data of peak patterns with the sequences in the plasmid blueprint/map. In other words, the use of prior probability (i.e. the low error rate during the plasmid construction in this case) can improve the posterior probability of correct base calling. Thus, high-quality sequencing data is not actually needed for the first ~700 bases, but researchers have had no choice but to obtain data with higher purity than necessary, because Sanger sequencing cannot balance data quality and cost (or read length). However, with nanopore sequencing, it is possible to accept a lower quality of data by reducing the number of reads per sample, with the benefit of lower cost. Here, we reduce this idea to practice, showing that the use of prior probability can prevent the quality of sequencing from falling too low, ensuring both low cost and high confidence of sequencing results. Using our approach, sufficient multiplexing is possible that the cost of whole-plasmid sequencing can drop to below that of a single Sanger sequencing run.

The outputs of SAVEMONEY are two FASTQ files: one that uses prior probability of the error during the plasmid construction at the last step of the post-analysis and one that does not. If a high number of reads is obtained, it does not matter which output file is used because they will return the same consensus. The problem occurs when the number of reads is low and these two consensus sequences do not match. Simulations in *Figure 6e* show that 'with prior' has lower error rate than 'without prior', whereas 'with wrong prior' has higher error rate. This trend indicates that the use of prior probability increased the specificity at the expense of sensitivity in the context of detecting errors (i.e. bases that differ between the blueprint and the actual plasmid). There is no clear standard for the extent to which 'with prior' results should be trusted, just as there is no clear standard for the manual inspection of an ambiguous position in a chromatogram of Sanger sequencing. However, even if an error is detected in 'without prior', there is a high probability that there is no error if indicated by the 'with prior' result. Therefore, in this instance, it would be advisable to re-sequence the plasmid without discarding it, as it is not fully rigorous to claim the absolute correctness of either the 'with prior' results indicating that there is an error or the 'without prior' results indicating that there is no error. This process is similar to what is typically done in Sanger sequencing, where the decision of whether to accept the sample, discard it, or sequence it again can be made when manually examining ambiguous regions of Sanger

sequencing chromatograms. The two outputs of SAVEMONEY allow for flexible data interpretation, which has been difficult to achieve with conventional analysis of nanopore sequence outputs.

SAVEMONEY can also be beneficial for those who own nanopore sequencers and perform library preparation, because SAVEMONEY can be executed to further de-multiplex the data after performing standard barcode-based sequencing. More plasmid variants than the number of available barcodes can in principle be sequenced simultaneously. Alternatively, in some cases, SAVEMONEY may even allow skipping the barcode introduction step entirely. One concern with using SAVEMONEY for those who own their own sequencers is the computational speed. A much higher number of reads can be obtained from in-house sequencers compared to outsourcing plasmid sequencing services, where the number of reads per sample typically ranges from 200 to several hundred. The prior information of known mixtures of plasmids used in SAVEMONEY speeds up the classification of raw reads from each pore and the generation of a multiple sequence alignment, which would otherwise be inherently difficult tasks. Because of this feature of our algorithm, computational cost is low, and the process can be executed even on a consumer-grade laptop computer. For example, in a case where five plasmids with sizes of 9999, 7958, 8627, 9325, and 10,107 bases were mixed and 653 reads were obtained from nanopore sequencing, it took 3.1 min to process deconvolution using MacBook Air (2020, Apple M1 processor: 8-core [4P+4E], 8-thread, 16 GB RAM). The same process was completed in 1.8 min using a mini gaming PC, Reatan Alloy 9 (2023, AMD Ryzen 9 7940HS: 8-core, 16-thread, 64 GB RAM). Nevertheless, to accommodate potentially heavier demands, we have implemented multiprocessing features to SAVEMONEY. The processing time is expected to be inversely proportional to the number of CPU cores used.

Other methods/algorithms for multiplexing plasmids have been proposed to reduce costs. For example, by using barcoded primers, pooled amplicons can be sequenced even if the original sequences are exactly the same (*Currin et al., 2019*). However, this approach requires additional primers, increases the number of procedures, and makes it impossible to sequence the entire plasmid. There is also an algorithm to mix plasmids in a barcode-free manner and unmix them *in silico* (*Mumm et al., 2023*). This algorithm is useful and proceeds at a similar speed to SAVEMONEY, processing the same dataset described above in 5.0 min, although it should be noted that the CPU and other specifications used in the web app server were not available. However, that approach is geared mainly toward users with medium-throughput sequencing capabilities, such as those who own sequencers and prepare libraries by themselves, and it is not clear whether it is suitable for end users who outsource sequencing to commercial services. In fact, as this approach does not use Bayesian analysis when obtaining consensus sequences to consider prior information, hundreds of reads per plasmid are needed for reliable sequencing. With a number this high, it is difficult to mix plasmids as extensively with SAVEMONEY when outsourcing to services such as Plasmidsaurus, which sometimes returns fewer than 200 reads per sample. In addition, the pipeline did not contain a pre-survey step, making it unclear for researchers to determine suitable combinations of plasmids to be mixed for outsourcing (*Mumm et al., 2022*). In contrast, we have shown that plasmids with differences of as little as two bases can be pooled and reliably de-mixed by SAVEMONEY with reliable consensus calculation using Bayesian analysis. We have also implemented a pre-survey step as part of the pipeline to guide researchers to find optimal combinations of plasmids to be mixed. This step greatly reduces the work involved in mixing plasmids and will help to spread nanopore sequencing as an attractive alternative to Sanger sequencing.

A limitation of our approach is that not all plasmids can be mixed. As described in the pre-survey algorithm section, plasmids from multiple colonies in the same plasmid construction procedure cannot be mixed because their expected sequences are identical. Nevertheless, SAVEMONEY can still save money if multiple plasmids are constructed simultaneously. For example, when sequencing two colonies from each of three different constructs (i.e. six plasmids in total), the standard approach would incur sequencing costs for six samples. However, with SAVEMONEY, up to three plasmids can be mixed per sample in this case, allowing all six plasmids to be sequenced as just two samples. As a result, the sequencing cost per plasmid is reduced by two-thirds. The strength of SAVEMONEY in this context is that plasmids that differ by as few as two bases can be mixed together. This number should be small enough to require minimal effort in finding mixable plasmids. Even among plasmids that are generally recognized as having high similarity, such as those used for single amino acid mutations, conferring resistance to RNAi, introducing different

peptide epitope tags, and constructing CRISPR guide RNAs, most of them should meet this criterion.

Another limitation of SAVEMONEY is that it calculates consensus base calling and quality scores independently for each nucleotide. However, the quality scores of neighboring bases are not independent in principle, because the number of bases producing current changes by moving through the pore is 5-mers (*Jain et al., 2015*). Going forward, incorporation of a model that can take base calling of neighboring nucleotides into account, such as a hidden Markov model, might further improve the quality of base calling (*Loman et al., 2015*).

There is still room for improvement in the maximum number of plasmids that can be mixed. Currently, the number of reads returned from the Plasmidsaurus service varies widely from hundreds to thousands, depending on the quality of the sample and/or variance of nanopore flow cells. If these unstable factors decrease and, for example, results with at least 1000 reads can be obtained every time, it will be possible to mix more than 30 plasmids, because we have calculated that the minimum required number of reads is 30 per plasmid. The number of plasmids that can be mixed together is expected to increase further with improvements to the base-level accuracy of nanopore sequencing technology. On the other hand, it is also challenging to prepare so many different plasmids. In practice, taking fully advantage of our algorithm might involve coordination between multiple colleagues in a lab who are constructing plasmids with different expected sequences. By enabling the mixing together of even a handful of plasmids, SAVEMONEY should dramatically drive down the costs for nanopore and other long-read sequencing technologies, further democratizing these powerful techniques for whole-plasmid and other long-read sequencing applications.

# Materials and methods

## Key resources table

| Reagent type (species) or resource | Designation | Source or reference | Identifiers | Additional information |
|---|---|---|---|---|
| Recombinant DNA reagent | pCDH1-lyn10-mCherry-LOVPLD* (plasmid) | This paper | | P1 in *Figure 2*; Plasmid used in PMID:38559292 with silent mutation |
| Recombinant DNA reagent | pCDH1-lyn10-mCherry-LOVPLD (plasmid) | PMID:38559292 | | P2 in *Figure 2* |
| Recombinant DNA reagent | pCDH1-lyn10-mCherry-LOVPLD** (plasmid) | This paper | | P3 in *Figure 2*; Plasmid used in PMID:38559292 with silent mutation |
| Recombinant DNA reagent | pCDH1-lyn10-mCherry-LOVPLD*** (plasmid) | This paper | | P4 in *Figure 2*; Plasmid used in PMID:38559292 with silent mutation |
| Recombinant DNA reagent | pET-17b-mod2_6xHis-PLDs48-HiBiT (plasmid) | This paper | | P5 in *Figure 2* |
| Recombinant DNA reagent | pET-17b-mod2_6xHis-PLD-HiBiT (plasmid) | This paper | | P6 in *Figure 2* |
| Recombinant DNA reagent | pET-17b-mod2_6xHis-PLDs4-HiBiT (plasmid) | This paper | | P7 in *Figure 2* |
| Recombinant DNA reagent | pET-17b-mod_6xHis-PLDs48**-HiBiT (plasmid) | This paper | | P8 in *Figure 2* |
| Recombinant DNA reagent | pET-17b-mod_6xHis-PLDs48*-HiBiT (plasmid) | This paper | | P9 in *Figure 2* |
| Recombinant DNA reagent | pET-17b-mod_6xHis-PLDs48-HiBiT (plasmid) | This paper | | P10 in *Figure 2* |
| Recombinant DNA reagent | pET-17b-mod_6xHis-PLDs27L484F-HiBiT (plasmid) | This paper | | P11 in *Figure 2* |
| Recombinant DNA reagent | pET-17b-mod_6xHis-PLD-HiBiT (plasmid) | This paper | | P12 in *Figure 2* |
| Recombinant DNA reagent | pET-17b-mod_6xHis-PLDs4A326T-HiBiT (plasmid) | This paper | | P13 in *Figure 2*, P1 in *Figure 3* |

*Continued on next page*

*Continued*

| Reagent type (species) or resource | Designation | Source or reference | Identifiers | Additional information |
|---|---|---|---|---|
| Recombinant DNA reagent | pET-17b-mod_6xHis-PLDs4-HiBiT (plasmid) | This paper | | P14 in *Figure 2*, P2 in *Figure 3* |
| Recombinant DNA reagent | pmNeonGreen-N1 (plasmid) | Other | | P1 in *Figure 3*; a gift from the Lammerding Laboratory, Cornell University, Ithaca, NY, USA |
| Recombinant DNA reagent | mCherry-Spo20 (plasmid) | Other | | P2 in *Figure 3*; a gift from the Frohman Laboratory, Stony Brook University, Stony Brook, NY, USA |
| Recombinant DNA reagent | GFP-PASS (plasmid) | Other | | P3 in *Figure 3*; a gift from the Du Laboratory, The University of Texas Health Science Center at Houston, Houston, TX, USA |
| Recombinant DNA reagent | iRFP-PASS (plasmid) | PMID:31999306 | | P4 in *Figure 3* |
| Recombinant DNA reagent | pcDNA3_P18-CIBN-P2A-CRY2-mCherry-PLD(1-17) (plasmid) | PMID:37217787 | | P5 in *Figure 3* |
| Recombinant DNA reagent | pcDNA3_CRY2-mCherry-PLD(2-27)-P2A-CIBN-CAAX (plasmid) | PMID:37217787 | | P6 in *Figure 3* |
| Recombinant DNA reagent | PLD-mCherry-Rab7 (plasmid) | Other | | P3 in *Figure 3*; constructed by Reika Tei (Baskin Lab) |
| Recombinant DNA reagent | dPLD-mCherry-Rab7 (plasmid) | Other | | P4 in *Figure 3*; constructed by Reika Tei (Baskin Lab) |
| Recombinant DNA reagent | pGFPN1-PL5(143–271)-EGFP-S161D | PMID:39209962 | | P5 in *Figure 3* |
| Recombinant DNA reagent | pGFPN1-PL5(143–271)-EGFP | PMID:35952650 | | P6 in *Figure 3* |
| Software, algorithm | SAVEMONEY | This paper | | version 0.3.4 |

## Plasmid preparation and sequencing

All plasmids were either purchased or constructed by Gibson Assembly (*Gibson et al., 2009*) or cut and paste cloning techniques. Restriction enzyme digestion tests were performed for those constructed before submitting to sequencing. All sequencing was performed by the Plasmidsaurus service certified by Oxford Nanopore Sequencing Technology. Sequencing data were obtained using V14 chemistry on PromethION with an R10.4.1 flow cell, followed by base calling using a high accuracy (HAC) model.

## Software packages used in SAVEMONEY scripts

All analyses were performed by using Google Colab or local environment (Python 3.10.0). The following packages were used: BioPython (*Chapman and Chang, 2000*), kpLogo (*Wu and Bartel, 2017*), Numpy (*Harris et al., 2020*), Pandas (*pandas-dev, 2024*), parasail (*Daily, 2016*), Pillow (*python-pillow, 2025*), PuLP (*Peschiera and Mitchell, 2024*), Spoa (*Vaser et al., 2017*), tqdm (*da Costa-Luis, 2019*), Scipy (*Virtanen et al., 2020*), and SnapGene Reader (*Luo, 2018*).

## Acknowledgements

This work was supported by the National Institutes of Health (R01GM143367 to JMB). MU was supported by an Overseas Research Fellowship from the Japan Society for the Promotion of Science and a Long-Term Fellowship from the Human Frontiers Science Program. We thank Jan Lammerding, Michael Frohman, Guangwei Du, Reika Tei, Xiaofu Cao, Shiying Huang, Po-Hsun Brian Chen, and Julia Li for providing their plasmids and sequencing data, as well as Saori Uematsu for providing benchmark data. We also thank Haiyuan Yu for helpful discussions.

## Additional information

### Funding

| Funder | Grant reference number | Author |
| --- | --- | --- |
| National Institutes of Health | R01GM143367 | Jeremy M Baskin |
| Japan Society for the Promotion of Science | Overseas Research Fellowship | Masaaki Uematsu |
| Human Frontier Science Program | Long-Term Fellowship | Masaaki Uematsu |

The funders had no role in study design, data collection and interpretation, or the decision to submit the work for publication.

### Author contributions

Masaaki Uematsu, Conceptualization, Software, Formal analysis, Funding acquisition, Validation, Investigation, Methodology, Writing – original draft, Writing – review and editing; Jeremy M Baskin, Supervision, Funding acquisition, Writing – original draft, Project administration, Writing – review and editing

### Author ORCIDs

Masaaki Uematsu ⓘ https://orcid.org/0000-0002-0197-8401
Jeremy M Baskin ⓘ https://orcid.org/0000-0003-2939-3138

Reviewer #1 (Public review): https://doi.org/10.7554/eLife.88794.3.sa1
Reviewer #2 (Public review): https://doi.org/10.7554/eLife.88794.3.sa2
Author response https://doi.org/10.7554/eLife.88794.3.sa3

## Additional files

### Supplementary files

Supplementary file 1. All sequences of plasmids used in this study.

MDAR checklist

### Data availability

SAVEMONEY is available through Google Colab. Locally executable scripts are available on GitHub (*Uematsu, 2024*) and on PyPI. The sequences of plasmids used in this study are available in *Supplementary file 1*.

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
