## [Editor Report · eLife Assessment]

This study provides an **important** computational tool for analyzing and deconvoluting a pool of plasmids sequenced without barcoding using nanopore long-read sequencing. The tool, which has been **convincingly** validated, is readily available to scientists interested in rapid and cost-effective verification of plasmid sequences as well as in scaling up analysis by pooling samples within barcodes.

---

## [Referee Report · Reviewer #1 (Public review)]

This manuscript presents SAVEMONEY, a computational tool designed to enhance the utilization of Oxford Nanopore Technologies (ONT) long-read sequencing for the design and analysis of plasmid sequencing experiments. In the past few years, with the improvement in both sequencing length and accuracy, ONT sequencing is being rapidly extended to almost all omics analyses which are dominated by short-read sequencing (e.g., Illumina). However, relatively higher sequencing errors of long-read sequencing techniques including PacBio and ONT is still a major obstacle for plasmid/clone-based sequencing service that aims to achieve single base/nucleotide accuracy. This work provides a guideline for sequencing multiple plasmids together using the same ONT run without molecular barcoding, followed by data deconvolution. The whole algorithm framework is well-designed, and some real data and simulation data are utilized to support the conclusions. The tool SAVEMONEY is proposed to target users who have their own ONT sequencers and perform library preparation and sequencing by themselves, rather than relying on commercial services. As we know and discussed by the authors, in the real world, to ensure accuracy, the researchers will routinely pick up multiple colonies in the same plasmid construction and submit for Sanger sequencing. However, SAVEMONEY is not able to support the simultaneous analysis of multiple colonies in the same run, as compared to the barcoding-based approaches. This is a major limitation in the significance of this work. Encouraging computational efforts in ONT data debarcoding for mixed-plasmid or even single-cell sequencing would be more valuable in the field.

Comments on revisions:

My previous concerns have been addressed, and the revised manuscript has been significantly approved.

---

## [Referee Report · Reviewer #2 (Public review)]

The authors developed an algorithm that allows to deconvolute plasmid sequences from a mixture of plasmids that have been sequenced by nanopore long read technology. As library preparations and barcoding of individual samples increases sequencing costs, the algorithm bypasses this need and thus decreases time on sample prep and sequencing costs. In a first step, the tool assesses which of the plasmid constructions can be mixed in a single library preparation by calculating a distance matrix between the reference plasmid and the constructions producing sequence clusters. The user is given groups of plasmids, from different clusters, to be pooled together for sequencing. After sequencing, the algorithm deconvolutes the reads by classifying them based on alignments to the reference sequence. A Bayesian analysis approach is used to obtain a consensus sequence and quality scores.

Strengths

The authors exploit one of the main advantages of long read sequencing that is to accurately resolve regions of high complexity, as regularly found in plasmids, and developed a tool that can validate plasmid constructions by reducing sequencing costs. Multiple plasmids (up to six) can be analyzed simultaneously in a single library without the need of sample barcoding, also reducing sample preparation time. Although inserts must be different, just 2 bases difference would be enough for correct assignation. Maximizes cost-efficiency for projects that require large amounts of plasmid constructions and high-throughput validation. The algorithm also allows for linear DNA analysis offering extra flexibility.

---

## [Author Response]

The following is the authors’ response to the original reviews

**Public Reviews:**

**Reviewer #1 (public review):**
This manuscript presents SAVEMONEY, a computational tool designed to enhance the utilization of Oxford Nanopore Technologies (ONT) long-read sequencing for the design and analysis of plasmid sequencing experiments. In the past few years, with the improvement in both sequencing length and accuracy, ONT sequencing is being rapidly extended to almost all omics analyses which are dominated by short-read sequencing (e.g., Illumina). However, relatively higher sequencing errors of long-read sequencing techniques including PacBio and ONT is still a major obstacle for plasmid/clone-based sequencing service that aims to achieve single base/nucleotide accuracy. This work provides a guideline for sequencing multiple plasmids together using the same ONT run without molecular barcoding, followed by data deconvolution. The whole algorithm framework is well-designed, and some real data and simulation data are utilized to support the conclusions. The tool SAVEMONEY is proposed to target users who have their own ONT sequencers and perform library preparation and sequencing by themselves, rather than relying on commercial services. As we know and discussed by the authors, in the real world, to ensure accuracy, the researchers will routinely pick up multiple colonies in the same plasmid construction and submit for Sanger sequencing. However, SAVEMONEY is not able to support the simultaneous analysis of multiple colonies in the same run, as compared to the barcoding-based approaches. This is a major limitation in the significance of this work. Encouraging computational ePorts in ONT data debarcoding for mixed-plasmid or even single-cell sequencing would be more valuable in the field.

We thank the reviewer for the positive response to our manuscript and the helpful comments.

The tool SAVEMONEY is proposed to target users who have their own ONT sequencers and perform library preparation and sequencing by themselves, rather than relying on commercial services.

We apologize that we were not clear enough in the manuscript. Our tool is designed for users who rely on commercial services (i.e., those who cannot include a barcode by themselves). However, it can also benefit those performing library preparation, as SAVEMONEY can be applied after standard barcode-based sequencing and de-multiplexing. The combination of standard barcodes with SAVEMONEY would significantly expands the scope of sequencing applications. For example, it would enable sequencing of more plasmid types than the number of available barcodes and, in some cases, it may even eliminate the need for barcode introduction. Because we do not own ONT equipment and because the primary target audience for the SAVEMONEY algorithm are users without ONT equipment, we were not able to conduct experiments using ONT. However, to clarify these possibilities, we added a dedicated paragraph describing these issues (3rd paragraph in the discussion section).

However, SAVEMONEY is not able to support the simultaneous analysis of multiple colonies in the same run, as compared to the barcoding-based approaches.

We agree with the reviewer about this limitation of SAVEMONEY, as it does not allow mixing of plasmids from multiple colonies in the same cloning run. However, that does not necessarily mean that SAVEMONEY cannot reduce sequencing costs in cloning. For example, when sequencing two colonies from each of three diPerent constructs (six plasmids in total), the standard approach would require sequencing costs for six samples. However, with SAVEMONEY, up to three plasmids can be mixed per sample, allowing them to be sequenced as just two samples. As a result, the sequencing cost per plasmid is reduced to one-third. The greatest benefits can be realized when SAVEMONEY is used at the laboratory level or by multiple researchers. To make this point clearer, we have added sentences in the 5th paragraph of the discussion section.

(1) To provide more comprehensive information for users who care about the cost, the Introduction section should include a cost comparison between Sanger and ONT, with more details, such as diPerent ONT platforms (MinION, PromethION, FlongIe), chemistries (flow cells) and kits. This additional information will be more helpful and informative for the users who have their own sequencers and are the target audience for SAVEMONEY.

We thank the reviewer for pointing this out. Since we do not own ONT equipment, we are unable to provide a total cost for using the ONT platform. However, we have included the price per sample (~$15 per plasmid) for the commercial service we have used, as well as the equipment that they employ (V14 chemistry on a PromethION with an R10.4.1 flow cell) and the number of reads obtained per plasmid (~100–1000) in the 4th paragraph of the introduction section. Though these costs will inevitably change over time, this information should still be helpful for those who own ONT sequencers in estimating the costs.

(2) In "Overview of the algorithm" (Pages 3-4) under the Results section, instead of stating "However, coverage varies from ~100-1000 and is diPicult to predict because each nanopore flow cell has diPerent properties.", it will be beneficial to provide more detailed information, such as sequencing length, yield/read count per flow cell of diPerent platforms. This information will assist users in designing their own experiments ePectively.

We thank the reviewer for the comment. As mentioned in the previous response, we are unable to provide sequencing length, yield/read count per flow cell because we do not own ONT equipment. However, we apologize if it was not clear in "Overview of the algorithm" section that we are discussing the use of results obtained from commercial services, and therefore we need to provide more detailed information about the results from the commercial service. We have now clarified in the sentence pointed out by the reviewr that the numbers are derived from the information provided by commercial sequencing services. In addition, we have also added that typical examples of the result properties, i.e., read length and quality score distribution, can be found in Fig. 2 at the end of the same paragraph.

(3) While this study optimized and evaluated the tool using a total of 14 plasmids, it may not provide suPicient power to represent the diversity of the plasmid world. Consideration should be given to expanding the dataset to include a broader range of plasmids in future studies to enhance the robustness and generalizability of the tool.

We are grateful to the reviewer for their valuable input. It is very reasonable that we had to expect that a larger number of plasmids should be used, even though the main target of SAVEMONEY is those who utilize commercial services. In the previous version of SAVEMONEY, it was not possible to process in a reasonable amount of time if too many plasmids were provided, though the algorithm itself does not have no restrictions based on the number of plasmids. Therefore, we have changed the underlying code to improve the algorithm, making it more than 20 times faster than the previous version (the benchmark time mentioned in the 3rd paragraph of the discussion section was improved to 3.1 minutes from the previous 65 minutes, using the same dataset and the same computer). Additionally, SAVEMONEY is now compatible with multiprocessing. The processing time is expected to decrease approximately inversely proportional to the number of CPU cores used. We have added these updates at the end of the 3rd paragraph in the discussion section.

(4) If applicable and feasible, including a comparison or benchmark of SAVEMONEY against other similar tools would further strengthen the manuscript. This comparison would allow users to evaluate the advantages and disadvantages of diPerent tools for their specific needs.

We thank the reviewer for the suggestion. We have added the benchmark using the similar tool, On-Ramp, with the exact same set of plasmids and FASTQ data used for our benchmark (4th paragraph in the discussion section). Because the machine specifications used in the On-Ramp web server are unknown, a direct comparison is not possible. However, using only laptop-level computational resources, SAVEMONEY was able to process the data 38% faster than On-Ramp. When using mini-PC level computational resources, the processing time was 64% faster than on-RAMP.

(5) The importance of pre-filtering raw sequencing reads should be emphasized as noisy reads can significantly impact the overall performance of the tool. It is essential to clarify whether any pre-filtering steps were performed in this study, such as filtering based on quality scores, read length, or other relevant factors.

We apologize for not being clear. Unfortunately, the commercial sequencing service we used did not provide the information regarding pre-filtering. However, the impact of the quality of pre-filtering based on quality score and read length on the quality of the final results is theoretically minimal in SAVEMONEY. First, during the initial step of the post-analysis, the classification step, short reads compared to the full plasmid length can be excluded based on the user-defined “score_threshold”. Simultaneously, low-quality reads with poor alignment to the plasmid can also be excluded, because “score_threshold” is related to the normalized alignment score. Even if there are low-quality reads that are not excluded at this stage, the ePect can be minimized during the final step of the post-analysis that generates consensus sequences. This is because our Bayesian analysis considers not only the base calling but also the q-scores to determine the consensus. Therefore, we believe the overall impact of pre-filtering on the final results is negligible.

(6) The statement regarding the number of required reads per plasmid (20-30) and the maximum number of plasmids (up to six) that can be mixed in a single run may become outdated due to the rapid advancements in ONT technology. In the Discussion section, instead of assuming specific numbers, it would be more beneficial to provide information based on the current state of ONT sequencing, such as the number of reads per MinION flow cell that can be produced.

We thank the reviewer for pointing this out. Because the number of required reads per plasmid depends on the accuracy of each read (i.e., the number of required reads can be reduced if the accuracy increases), we have added the description of these points to the last paragraph of the discussion section.

**Reviewer #2 (public review):**
The authors developed an algorithm that allows for deconvoluting of plasmid sequences from a mixture of plasmids that have been sequenced by nanopore long read technology. As library preparations and barcoding of individual samples increase sequencing costs, the algorithm bypasses this need and thus decreases time on sample prep and sequencing costs. In the first step, the tool assesses which of the plasmid constructions can be mixed in a single library preparation by calculating a distance matrix between the reference plasmid and the constructions producing sequence clusters. The user is given groups of plasmids, from diPerent clusters, to be pooled together for sequencing. After sequencing, the algorithm deconvolutes the reads by classifying them based on alignments to the reference sequence. A Bayesian analysis approach is used to obtain a consensus sequence and quality scores.StrengthsThe authors exploit one of the main advantages of long-read sequencing which is to accurately resolve regions of high complexity, as regularly found in plasmids, and developed a tool that can validate plasmid constructions by reducing sequencing costs. Multiple plasmids (up to six) can be analyzed simultaneously in a single library without the need for sample barcoding, also reducing sample preparation time. Although inserts must be diPerent, just 2 bases diPerence would be enough for a correct assignation. It maximizes cost-ePiciency for projects that require large amounts of plasmid constructions and highthroughput validation.

We thank the reviewer for the positive response to our manuscript and the helpful comments.

WeaknessesThe method proposed by the authors requires prior knowledge of plasmid sequences (i.e., blueprints or plasmid reference) and is not suitable for small experiments. The plasmid inserts or backbones must be diPerent e.g., multiple colonies from the same plasmid construction ePort cannot be submitted together.

As also discussed in the response to reviewer 1, we agree with the reviewer that SAVEMONEY does not allow you the analysis of plasmids from multiple colonies in the same cloning experiment. However, that does not necessarily mean that SAVEMONEY cannot reduce the sequencing cost. For example, when sequencing two colonies from each of three diPerent constructs (six plasmids in total), the standard approach would require sequencing costs for six samples. However, with SAVEMONEY, up to three plasmids can be mixed per sample, allowing them to be sequenced as just two samples. As a result, the sequencing cost per plasmid is reduced to one-third. The greatest benefits can be realized when SAVEMONEY is used at the laboratory level or by multiple researchers. To make this point clearer, we have added sentences in the 5th paragraph of the discussion section.

The reviewer also expressed concern that SAVEMONEY is not suitable for experiments at a small scale. To put it more precisely, SAVEMONEY cannot be used when the experiment size is minimal, such as in a lab that consistently constructs only a single plasmid at a time. That said, the strength of SAVEMONEY lies in its scalability. Even in labs where plasmid construction is typically limited to one at a time, there may be occasional instances where two or more plasmids are created simultaneously. In such cases, SAVEMONEY can be used to reduce sequencing costs. Moreover, in a typical molecular biology lab where multiple plasmids are constructed every week, SAVEMONEY can be particularly ePective. Given its adaptability and cost-saving potential and widespread use since its initial publication on *bioRxiv* and on Google Colab, we are confident that SAVEMONEY will continue to be a valuable tool for a wide range of researchers.

**Recommendations For The Authors:**

**Reviewer #2 (Recommendations For The Authors):**
The manucript assumes all samples are sent out for sequencing at a specific company. This could be generalized for a much broader use since many labs now own nanopore sequencers. In turn, the advantage of reducing hands-on sample prep becomes more evident.

We thank the reviewer for pointing this out. We agree that SAVEMONEY can also benefit those performing library preparation. Combination of standard barcodes with SAVEMONEY significantly expands the scope of sequencing applications. For example, it enables sequencing of more plasmid types than the number of available barcodes and, in some cases, may even eliminate the need for the sample prep step to introduce barcode. Because we do not own ONT equipment, we could not conduct experiments using ONT. However, to clarify these possibilities, we added a dedicated paragraph (3rd paragraph in the discussion section).

The base calling model (high accuracy, super accuracy) used by Plasmidsaurus and tested here should be mentioned.

We thank the reviewer for the suggestion. The description about the base calling model (HAC) was added in Materials and Methods section.

Other modifications to the revised manuscript

Beyond changes made in response to reviewer comments above, we have also through our continued use and improvement of SAVEMONEY, made additional changes to the algorithm and therefore to the manuscript. Those changes are outlined below. Improvements in the pre-survey step

(1) The pre-survey algorithm was reduced to a Zero-One Integer Linear Programming Problem to guarantee the optimal combinations, as previous versions did not ensure an optimal solution. Relatedly, the explanation of the algorithm in the main manuscript was updated.

(2) The algorithm was modified to ensure that the number of plasmids distributed to each group is balanced. A new feature was also added to allow users to specify the number of groups, which is beneficial when balancing between cost and quality.

(3) An error was corrected in Fig. 2, where the distance calculation method for the hierarchical clustering step for group formation was Farthest Point Algorithm, which calculates distance between two clusters based on the farthest pair of plasmids. The correct method is the Nearest Point Algorithm. This error was present only in Fig. 2, while other implementations, including source code of SAVEMONEY and Google Colab page, were correct from the beginning. We have corrected the error in Fig. 2.

Modifications in figures, manuscripts, and other aspects

(1) Fig. 3 was updated to reflect the update of SAVEMONEY, although it did not show any important diPerences.

(2) Parameter names were updated as follows:

“threshold (pre)” -> “distance_threshold”

“threshold (post)” -> “score_threshold” Added “number_of_groups”

(3) The order of elements was rearranged in Fig. 4.

(4) Incorrect calculations were fixed in Fig. 4g, h, and i (old Fig. 4d, h, and l). Related to that, Fig. 4j, k, and l and Table 1 were added, in addition to the explanation in the main manuscript.

(5) SAVEMONEY was packaged and was released on PyPI to facilitate easy installation and integration by other developers.

(6) SAVEMONEY was updated and expanded to accommodate linear DNA fragments, such as PCR amplicons and long synthetic DNA. Users can select the topology of DNA by specifying that as an option. A description of this new capability was added at the end of “Overview of the algorithm” section.